# Detecting and Tracking Significant Events for Individuals on Twitter by Monitoring the Evolution of Twitter Followership Networks

**Tao Tang [1] and Guangmin Hu [2],***

[1] School of Information and Communication Engineering, University of Electronic Science and Technology of China, Chengdu 611731, China; terotongcn@gmail.com

[2] School of Resources and Environment, University of Electronic Science and Technology of China, Chengdu 611731, China

* Correspondence: hgm@uestc.edu.cn; Tel.: +86-028-6183-0209

**Abstract:** People publish tweets on Twitter to share everything from global news to their daily life. Abundant user-generated content makes Twitter become one of the major channels for people to obtain information about real-world events. Event detection techniques help to extract events from massive amounts of Twitter data. However, most existing techniques are based on Twitter information streams, which contain plenty of noise and polluted content that would affect the accuracy of the detecting result. In this article, we present an event discovery method based on the change of the user's followers, which can detect the occurrences of significant events relevant to the particular user. We divide these events into categories according to the positive or negative effect on the specific user. Further, we observe the evolution of individuals' followership networks and analyze the dynamics of networks. The results show that events have different effects on the evolution of different features of Twitter followership networks. Our findings may play an important role for realizing how patterns of social interaction are impacted by events and can be applied in fields such as public opinion monitoring, disaster warning, crisis management, and intelligent decision making.

**Keywords:** ego network; events; network dynamics; Twitter

## 1. Introduction

Twitter is one of the most popular online social media platforms with more than 300 million monthly active users. Users are allowed to publish short messages no more than 140 characters called tweets to express what is happening in the world and their opinions about it. A user can build directed social connections to others representing the "follow" relationship. The initiator of a connection is called the follower, and the recipient is referred to as a followee or a friend of him. When a user posts some new content on Twitter, his followers can receive these messages on their own homepages and interact with him like forwarding (it's called retweet on Twitter) or replying to the tweets. Through subscribing to the content posted by others, a user can learn about what is happening with people and organizations he or she is interested in.

It has almost no barrier to post tweets on Twitter, people can share tweets about everything even trivial matters that traditional media won't pay attention to. Twitter is also famed for spreading messages almost instantly. For instance, an earthquake occurred in Morgan Hill, California on 30 March 2009. The first geocoded tweet about the earthquake arrived 19 s later, while the Northern California Seismic Network who has the Advanced National Seismic System cost 22 s to make an automatic response. Given all those factors, Twitter has become one of the main sources of acquiring information about real-life events. Twitter provides public Application Program Interfaces (APIs) for users to

capture data which of interest. Users can collect real-time Twitter feeds or short-term historical tweets with specific keywords through these APIs. Using event detection techniques on Twitter data streams can help us to know which events are happening or happened recently, then we can consider for further analysis like the importance of events or the spatial and temporal patterns. Commonly, people show concern about real-time hot events or specific types of events, but here we pay more attention to events which are significant for particular individuals. For instance, suppose you are a fan of a celebrity, then you would like to know what happened to the celebrity or which events he is concerned about. In this article, we refer to these events are Personal Important Events (PIEs) for an individual. Individuals might not be involved in his PIEs (be one of the main characters of the events) but just participated in the discussion about those events. PIEs have a considerable effect on an individual, and it reflects on the change of the individual's local network structure. For example, a user shared a good viewpoint on an event and endorsed by many other users. Then he might get numerous new followers on Twitter and thus his local network structure changed sharply.

The local networks of users are always in an evolutionary process since there will be constantly new follow and unfollow behaviors. In this article, we study the dynamics of personal local networks of two Twitter accounts and observe how they evolve. We found that Twitter followership networks are highly dynamic. Within the one and a half months of our observation, about 9% and 22% of all connections have changed in the two personal local networks respectively. Most of the time, the evolution of personal local networks is in a steady state that new follows and unfollows come at a stable rate. However, when a PIE occurs, the violent disturbance is produced. In other words, PIEs cause bursts in the dynamics of the local network structure. We find that PIEs lead to two phenomena. One is that plenty of other users would link to the individual almost simultaneously, we refer to it as the follow burst. The other one is the individual's many followers drop the follow connections consecutively in a short time, and we call it the unfollow burst. Sometimes the two phenomena simultaneously occur on the individual. These bursts will significantly change the user's local network structure.

The remainder of this paper is organized as follows: We briefly review related works in Section 2. Then we describe our dataset and empirically study the evolution of users' personal social networks in Section 3. In Section 4 we introduce how to detect PIEs from user behaviors. We research on the effect of PIEs on users' local social network structure in Section 5 and give a simple sum-up in Section 6.

## 2. Related Works

Twitter event detection has always been a hot topic since Twitter launched in 2006. Researchers have shown that twitter event detection contributes to various application fields, such as epidemic diseases [1], political affairs [2–6], traffic conditions [7] and natural disaster emergencies [8–11]. Twitter event detection can be classified as specified and unspecified according to the event type.

Specified event detection uses pre-defined event information (e.g., keywords, hashtags) or known events. Lee and Sumuya [12] utilized the collective experiences and crowd behaviors on Twitter to detect geo-social events. Khurdiya et al. [13] proposed a framework based on Searching on Lucene with Replication (SOLR) and Conditional Random Field (CRF) which can identify small sub-events around a major event and build a map of them. Rill et al. [3] presented a system that uses special sentiment hashtags to detect emerging political events. Huang et al. [14] designed a high utility patter clustering framework that aims to detect and visualize small-scale city-level or even street-level events. Adedoyin–Olowe et al. [15] considered hashtags a significant and primary feature and used frequent pattern mining to capture word occurrence and detect sports and political events.

Unspecified event detection does not consider prior event information and mainly rely on bursty features, which is closer to our work. Abdelhaq et al. [16] developed a system called EvenTweet to detect localized events in real-time from a Twitter stream. Gao et al. [17] detected geographical social events by mining geographical temporal patterns and analyzing the content of tweets. Liu et al. [18] presented a system to detect burst events through mining burst words by incorporating features from

message content, propagation periods, and other characteristics. Zhou and Chen [19] proposed a framework called Variable Dimensional Extendible Hash (VDEH) which fully utilizes the information of social data over multiple dimensions to detect composite social events over streams. Cheng and Wicks [20] used Space-Time Scan Statistics (STSS) technique to detect significant space-time events without using of tweet context. Alsaedi et al. [21] proposed an online combined classification-clustering framework to identify real-time events. Zhang et al. [22] introduced a graph-based event detection technique where nodes in the graph represent burst words and the edge weight is calculated according to their co-occurrence within each tweet. Strongly connected components in the graph are identified through the graph clustering technique that uses the depth-first search algorithm and the connected components are considered as events. Zhou et al. [23] developed a Bayesian model-based framework called Latent Event and Category Model (LECM) which assumes that each tweet is associated with an event. LECM extracts events from tweets and groups them into clusters with event type labels.

These event detection techniques utilize various kinds of features extracted from Twitter data, but new follow/unfollow count is rarely used. Since our work aims to detect the occurrence of events and does not focus on the details of events, we put forward a simple and efficient method which use new follow/unfollow count to detect events.

Except for proposing a new Twitter event detection method, we also analyze the dynamics of individuals' local followership networks. In fact, many works that target exclusively to the dynamics of online social networks have been done. In the early period, the emphasis of the works is on the modeling of various aspects of social network evolution over time [24–26]. More recently, the research hotspot has shifted to predict local changes in the network, such as the addition and deletion of specific edges between social actors [27–30]. In the process of predicting the edge creation and deletion, many features were found to be helpful, such as the network topological structure [31–33], the internal influence among the social actors [34–36], the external influences like other forms of media [37,38], and the nature of the information content itself [39–42]. Our work does not focus on the relevance between network dynamics and particular features, but stands in a macro perspective and identifies how events affect the evolution of social networks.

## 3. Brief Analysis of Twitter Followership Graph

In the section, we introduce our dataset which consists of multiple snapshots of two users' local social networks. We briefly analyzed the change of followers count, and further analysis of more features of Twitter followership networks will be done later.

### 3.1. Dataset Description

Our analysis focus on the users' local social networks. To keep things simple, we define a user's ego network as the subgraph made up of a user's followers (excluding the user himself) and all the follow relationships between them. The user himself is known as the ego and his followers are called alters.

Our dataset includes snapshots of two Twitter ego networks. The egos we chose are *@NZNationalParty* and *@nzlabour*, who are the official Twitter accounts of New Zealand Nation Party and New Zealand Labour Party separately. We observe the dynamics of the two networks from 10 September 2017 to 24 October 2017. This period of time just catches up the New Zealand general election. We fetch the data of the two egos' followers via the Twitter Representational State Transfer (REST) API. Twitter modified the access permission of REST API in 2013 and imposed a limit of 15 times requests to get the lists of followers each 15 min window for an authenticated user. This rate-limiting makes it harder to get a complete picture of the Twitter social graph. That is an important reason of the shifting for the research emphasis from the global Twitter network to the local Twitter network. The two ego networks have 14,306 and 23,877 nodes at the beginning respectively. We have recorded the exact timestamps of connections that created or deleted during the observation, hence we can study the fine-grained network dynamics.

## 3.2. Highly Dynamic of Twitter Network

Go through the everyday snapshots, we find that the Twitter ego networks are highly dynamic. Compare with the starting time, about 6.6% new connections came up in ego-network1 and 2.39% old links were deleted. Ego-network2 added about 19.1% new edges during the period, while 2.8% old links were removed. This shows the highly dynamic nature of Twitter followership networks. The removal of old edges accounts for a respectable proportion, which demonstrates that the thought of considering the Twitter graph as an 'only-growing' network in some previous works [27] is incorrect.

Figures 1 and 2 plot the counts of the everyday new follows and unfollows of *@NZNationalParty* and *@nzlabour* respectively. From Figures 1 and 2, we can discover that the churn rate of the followers of an individual remains consistent at most times. This steady background volatility gets interrupted when events happen, and then the network structure changes significantly.

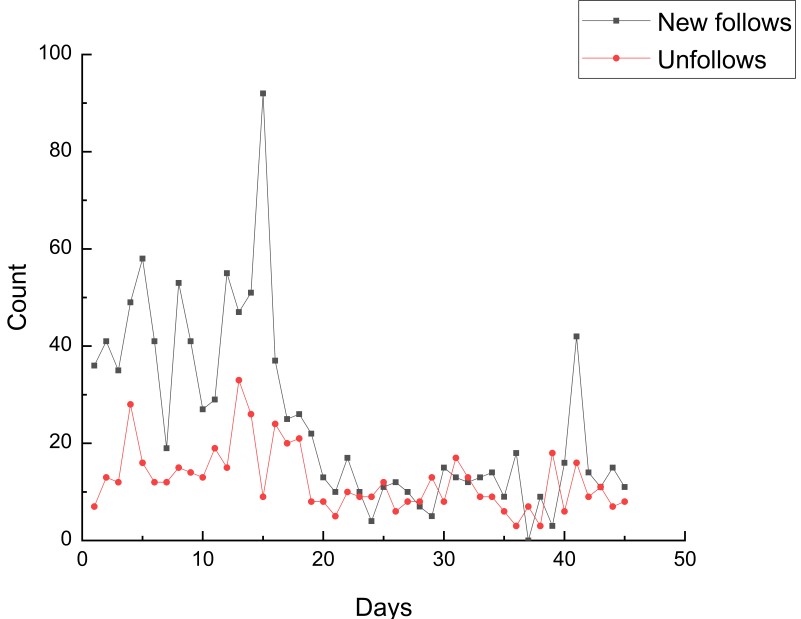

**Figure 1.** The variation of followers count of *@NZNationalParty*.

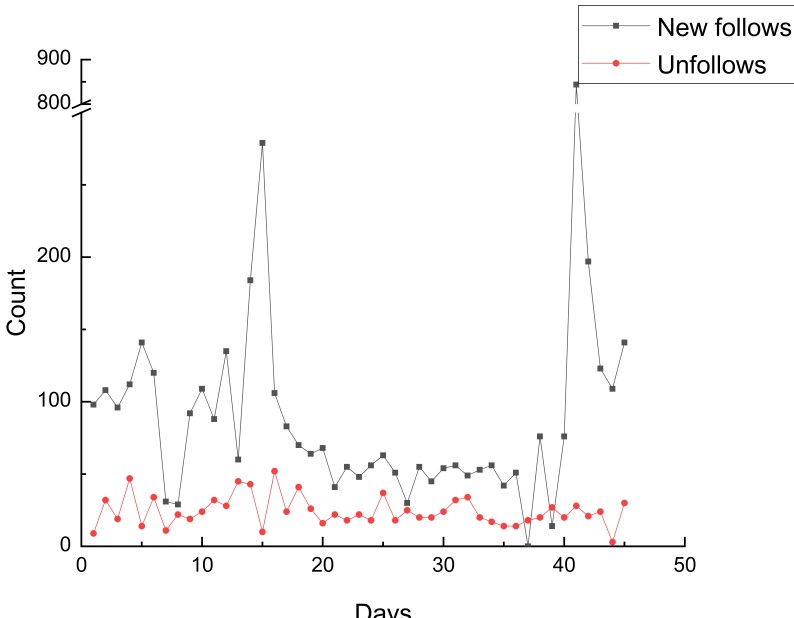

**Figure 2.** The variation of followers count of *@nzlabour*.

## 4. Personal Important Events Detection

### 4.1. Personal Important Events

A mass of events happens on Twitter every day, but most of them have few effects on a specific user's local network structure. Only if a user is referred to an event or participates in the discussion about the event, the event might provoke a great change of the users' local network structure. We define such events which significantly change a user's local network structure as Personal Important Events (PIEs) for the user.

We utilize the count variation of a user's followers to judge the PIEs for the users. That means, if a user experienced a follow burst or an unfollow burst in a time interval, we claim that a PIE for the user happened at that time. Then we try to discover PIEs of our observed objects, *@NZNationalParty* and *@nzlabour*. To detect the PIEs, or in other words, to identify the time intervals in which a user receives much more new follows or unfollows compared to what was expected historically, we process the data as follows.

### 4.2. The Bursts and the PIEs Detection

We use a method analogous to the way Myers and Leskovec used in [43] to detect bursts. We treat the arrival of new follows/unfollows of a user as an independent time series. We set the time interval as one day. Let $x = \{x_1, x_2, \ldots, x_n\}$ be the number of new follows/unfollows a user receives for each day. Let $t_i$ denote the $i^{th}$ day of our observation period, and let $f(t_i)$ represent the difference between the actual new follows/unfollows and the expected value during $t_i$:

$$f(t_i) = x_i - E[x \mid t_i] = x_i - \frac{\sum_{j; 0 < t_i - t_j \leq 2} x_j \cdot \omega(t_i - t_j)}{\sum_{j; 0 < t_i - t_j \leq 2} \omega(t_i - t_j)} \tag{1}$$

There $\omega(t_i)$ is an exponentially decaying weight function. When the value of $f(t_i)$ is greater than a threshold, we consider a PIE happens at day $t_i$. In this article, we set the threshold as three standard deviations of the time series according to PauTa Criterion ($3\sigma$ Criterion). We detected 5 PIEs for *@NZNationalParty* and 7 PIEs for *@nzlabour* during the observation period. We numbered the PIEs according to the time. As shown in Figure 3, the PIE 1~5 which are related to *@NZNationalParty* happened at Day4 (13 September 2017), Day12, Day14, Day15, and Day41 severally. Moreover, the PIE 6~12 related to *@nzlabour* happened at Day12, Day14, Day15, Day25, Day28, Day38, and Day41 separately. They are plotted on Figure 4. Refer to the news media, we find that PIEs are likely to correspond to real-world events about the New Zealand general election. We list the detected PIEs and the real-world events in Table 1 according to the dates.

**Table 1.** The correspondent relationship between Personal Important Events (PIEs) and real-world events.

| | Detected PIEs | Real-World Events |
|---|---|---|
| 09.13.2017 | PIE1 | The National Party denounced the tax policy of the Labour Party. |
| 09.21.2017 | PIE2, PIE6 | The final televised election debate was held. |
| 09.23.2017 | PIE3, PIE7 | The general election was held. |
| 09.24.2017 | PIE4, PIE8 | The preliminary result of electoral votes was announced. |
| 10.04.2017 | PIE9 | The First Party prepared to negotiate with the National Party and the Labour Party. |
| 10.07.2017 | PIE10 | The statistics for special votes was completed. |
| 10.17.2017 | PIE11 | The leader of the Labour Party was suspected to hint that she's gonna win. |
| 10.20.2017 | PIE5, PIE12 | The Labour Party won the election officially last night. |

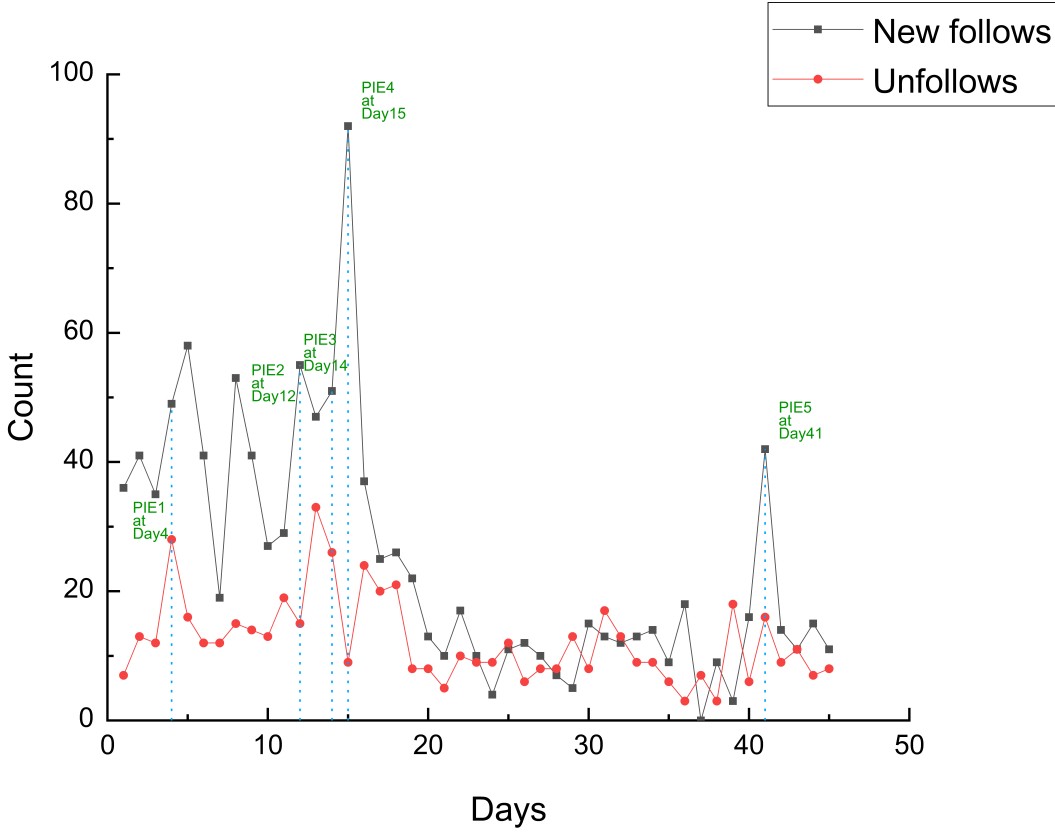

**Figure 3.** PIE 1–5 related to *@NZNationalParty*.

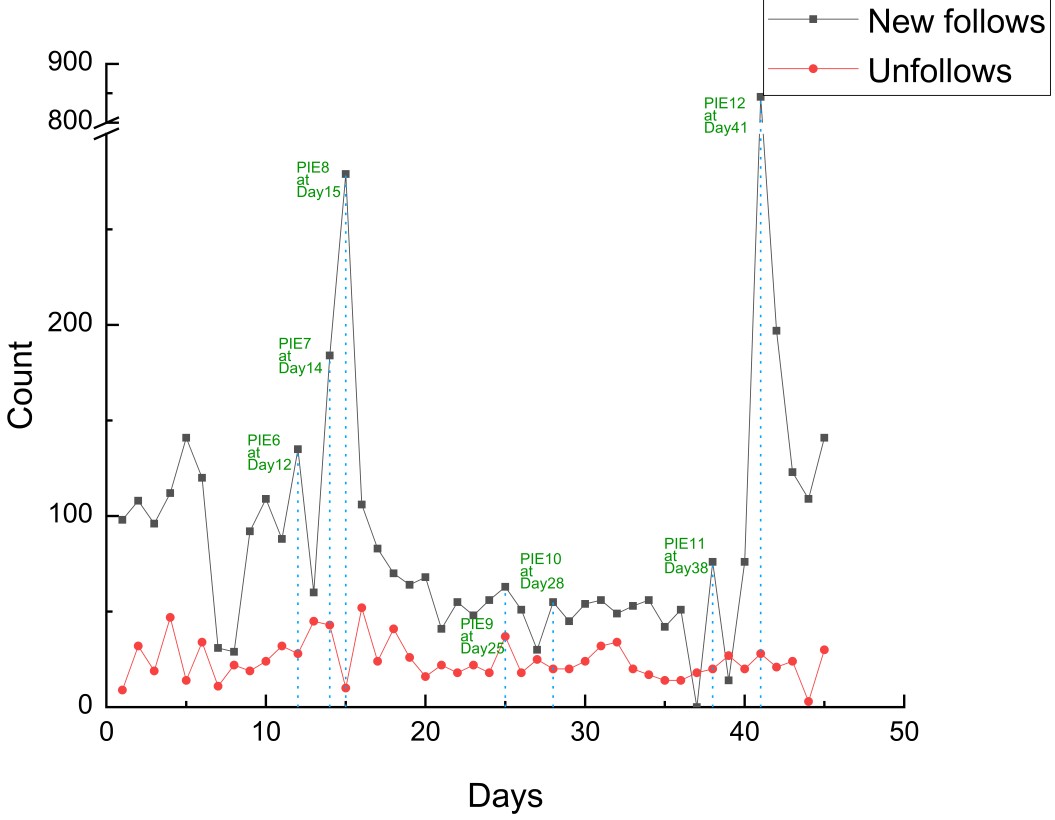

**Figure 4.** PIE 6–12 related to *@nzlabour*.

We classify PIEs into three categories according to the consequences brought to the corresponding user: follow-burst-PIEs for PIEs which lead to a follow burst, unfollow-burst-PIEs for PIEs which cause an unfollow burst, and mixed-burst-PIEs for PIEs result in both follow burst and unfollow burst simultaneously. Here we find that the PIE 1, 2, 4 for @*NZNationalParty* and the PIE 6, 8, 10 for @*nzlabour* are follow-burst-PIEs. The PIE 3 for @*NZNationalParty* and the PIE 7, 9 for @*nzlabour* are unfollow-burst-PIEs. The rest of PIEs including PIE 5, 11, 12 are mixed-burst-PIEs. The categories of PIEs are listed in Table 2. In a general way, a follow-burst-PIE for a given user is in favor of him in public opinion. On the contrary, an unfollow-burst-PIE for a user corresponds to an event which against him. The situation of a mixed-burst-PIE could be complicated since we are not sure whether a positive effect or a negative effect is brought to the related user.

**Table 2.** The categories of PIEs.

| Related User of PIEs | @*NZNationalParty* | @*nzlabour* |
|---|---|---|
| Follow-burst-PIEs | PIE1, PIE2, PIE4 | PIE6, PIE8, PIE10 |
| Unfollow-burst-PIEs | PIE3 | PIE7, PIE9 |
| Mixed-burst-PIEs | PIE5 | PIE11, PIE12 |

## 5. Evolution of Twitter Ego Networks

With the constant occurrence of link creation and deletion, the users' local network structure evolves as time goes by. In this section, we'll see how the two ego networks in our dataset evolve and how the bursts caused by PIEs effect on the evolution. Due to the complexity of mixed-burst-PIEs, here we only consider the effect of follow-burst-PIEs and unfollow-burst-PIEs on the ego network dynamics.

### 5.1. Follower Tweet Similarity

A user directly influences his followers (e.g., the members of his ego network) on Twitter through information diffusion. It's necessary to understand the relationship between a user and his followers. So we first explore how similar a user is to his followers, and how this similarity changes after the occurrence of a PIE for him. From a user's tweets (including retweets), we can basically know what he is interested in. For a pair of users, the textual similarity of their tweets would be a good indicator to quantify how similar their interests are. We aggregate the tweets posted by each user during the observation into single documents. Then we compute the cosine similarity of the Term Frequency-inverse Document Frequency (TF-IDF) weighted word vectors between two users' aggregated tweet documents and adopt it as user tweet similarity. First we extract all the key terms (assume that $m$ terms in total) $\mathcal{W} = \{w_1, w_2, ..., w_m\}$ from documents (of tweets). For a user $x$ and his document (of tweets) $d_x$. The term frequency $tf_{(w_1, d_x)}$ measures how frequency term $w_1$ occurs in document $d_x$:

$$tf_{(w_1, d_x)} = \frac{\text{Number of times term } w_1 \text{ appears in document } d_x}{\text{Total number of terms in document } d_x} \tag{2}$$

The inverse document frequency $idf_{(w_1)}$ measures how important term $w_1$ is in all the documents:

$$idf_{(w_1)} = \ln \frac{\text{Total number of documents}}{\text{Number of documents with term } w_1 \text{ in it}} \tag{3}$$

Then the TF-IDF weight of term $w_1$ in document $d_x$ is $tf\text{-}idf_{(w_1, d_x)} = tf_{(w_1, d_x)} \cdot idf_{(w_1)}$. Document $d_x$ can be represented by a $m$ dimension TF-IDF vector:

$$tf\text{-}idf_{(d_x)} = (tf\text{-}idf_{(w_1, d_x)}, tf\text{-}idf_{(w_2, d_x)}, ..., tf\text{-}idf_{(w_m, d_x)}) \tag{4}$$

In that way, the text similarity of user $x$ and user $y$ can be computed according to the cosine similarity between corresponding document TF-IDF vectors:

$$
\begin{aligned}
TextSim(x,y) &= \frac{tf\text{-}idf_{(d_x)} \cdot tf\text{-}idf_{(d_y)}}{\|tf\text{-}idf_{(d_x)}\| \cdot \|tf\text{-}idf_{(d_y)}\|} \\
&= \frac{\sum_{k=1}^{m} tf\text{-}idf_{(w_k,d_x)} \cdot tf\text{-}idf_{(w_k,d_y)}}{\sqrt{\sum_{k=1}^{m} tf\text{-}idf_{(w_k,d_x)}}^2 \cdot \sqrt{\sum_{k=1}^{m} tf\text{-}idf_{(w_k,d_y)}}^2}
\end{aligned}
\tag{5}
$$

We define the follower tweet similarity of a user as the average value of the tweet similarity between a user and all his followers. By observing the follower tweet similarity of a user before and after a PIE, we investigate whether user's followers become more similar in their interests. We measured the follow tweet similarity of the egos for multiple days before and after the PIEs. To make the metric comparable across different users, we normalize each measurement by its value exactly at the day of the PIE and then average the metric across all PIEs of the same type. Figure 5 shows the result averaged across all follow-burst-PIEs and unfollow-burst-PIEs.

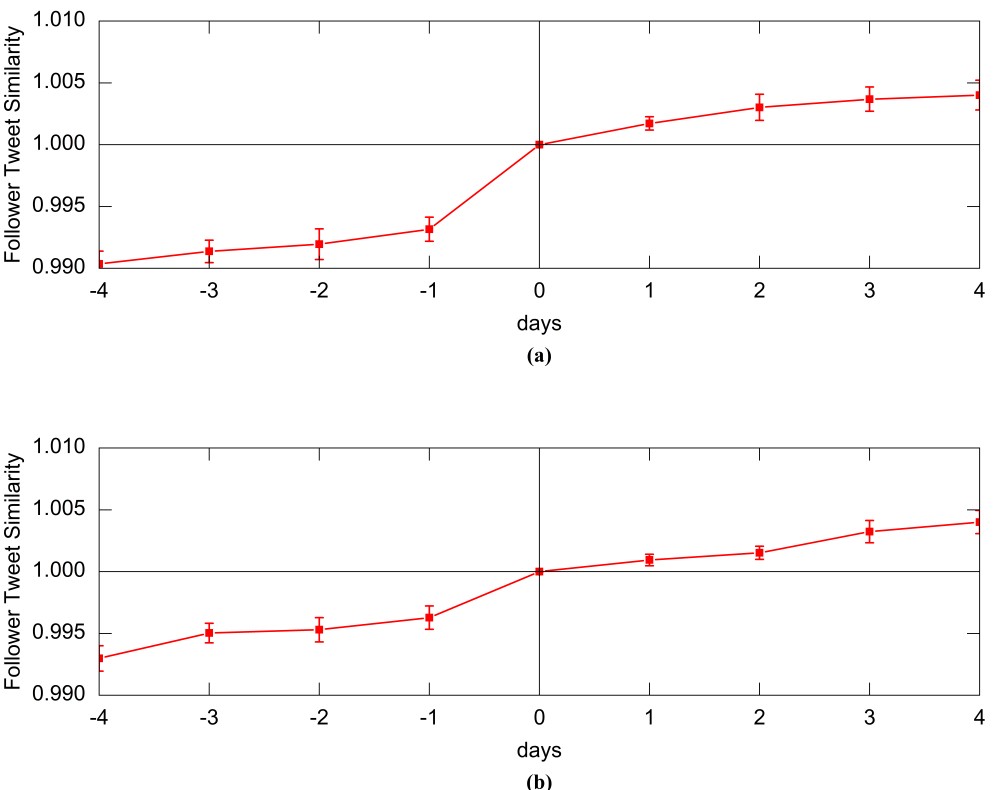

**Figure 5.** (**a**) Normalized Follower Tweet Similarity before and after a follow-burst-PIE; (**b**) Normalized Follower Tweet Similarity before and after an unfollow-burst-PIE.

We observe that the follower tweet similarity basically keeps increasing over time. This is easy to understand. When a user is new to Twitter, his earliest followers might not be that similar to him because they don't know his interests well enough. As the user's online behaviors increase, he is more likely to be followed by users who share common interests with him. We conclude that after whether a follow-burst-PIE or an unfollow-burst-PIE, the follower tweet similarity of ego networks obviously grows faster. A PIE makes the ego get more exposure on Twitter so users get to know more about his interests. Intuitively, a follow-burst-PIE attracts similar users to follow the ego, and an unfollow-burst-PIE prompt existing followers who are not similar to the ego to unfollow him. Both of them accelerate the rise of follower tweet similarity.

### 5.2. Follower Tweet Coherence

We discover that the followers become more in common with the ego after a PIE in the previous paragraph, then we are going to see if the followers become more related to one another (not just to the ego user). We use the same method of TF-IDF cosine similarity of tweet content to measure the similarity among the followers. Here we define the follower tweet coherence of a user as the average value of the tweet similarity of all pairs of his followers. We measure the tweet similarity across all pairs of followers of the ego in the days succeeding and preceding the PIEs. For large-scale networks who have too much node pairs, biased sampling strategies such as sample edge count [44] are recommended to calculate similarity between node pairs. The negative effect of sampling on similarity calculating is proved to be small [45]. These measurements are normalized by the value exactly on the day of the PIE. Moreover, we average the metric across all PIEs of the same type just as we did before. We plot the result in Figure 6.

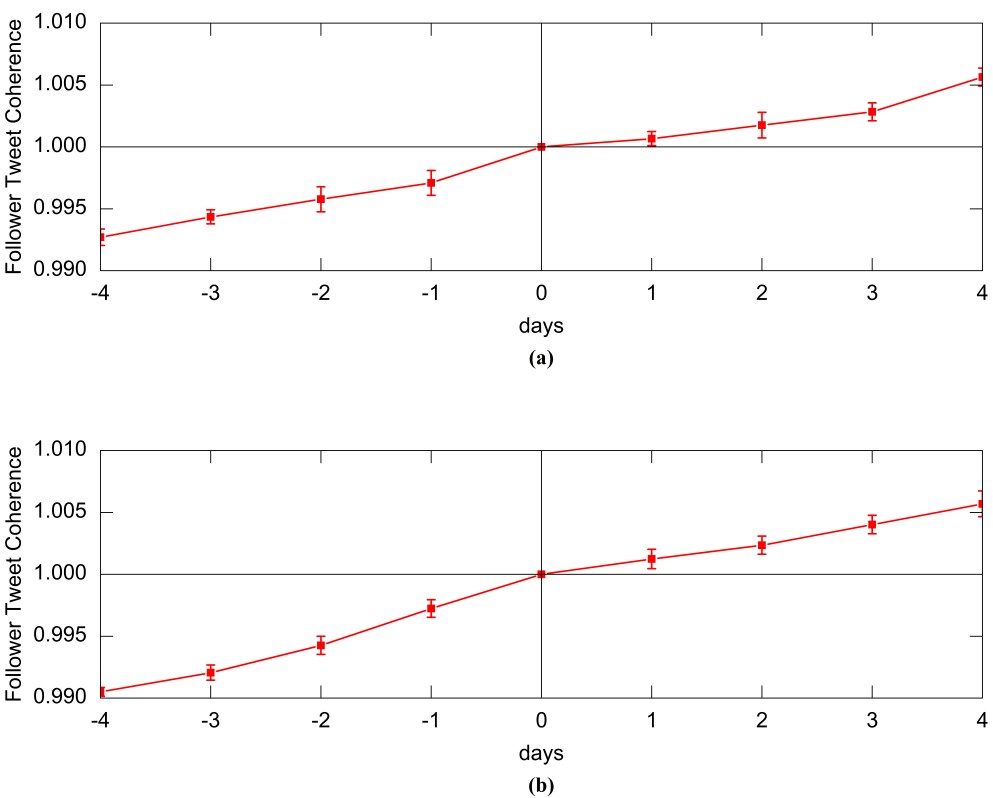

**Figure 6.** (**a**) Normalized Follower Tweet Coherence before and after a follow-burst-PIE; (**b**) Normalized Follower Tweet Coherence before and after an unfollow-burst-PIE.

A similar result to the follower tweet similarity is shown. Likewise, the follower tweet coherence increases over time, and speeds up after both follow-burst-PIEs and unfollow-burst-PIEs. Both types of PIEs cause the followers' interests and tweets to become more aligned with each other, make the user's ego network become more homogeneous. Since the followers become more similar to the ego as time goes by, it's predictable that followers become more alike among themselves. Combined with the result of follower tweet similarity, it's indicated that PIEs cause a process in the ego network's evolution toward bringing similar users together and pushing dissimilar users farther apart.

### 5.3. Connected Components Amongst Followers

After analyzing the similarity relationship between users before and after a PIE, the structural changes of a user's local neighborhood will be explored. In detail, we calculate the number of weakly connected components (WCC) of the ego networks during a burst. A weakly connected component of

a directed graph is a maximum subgraph in which any two nodes are connected by direct edge path. For any user in a weakly connected component of a Twitter followership network, there exists at least one another user follows or followed by him. If the number of WCC of an ego network is high, that means the subgraph of the user's followers is fragmented. It indicates that user's followers tend not to follow each other and do not belong to a single cohesive community. After measuring the number of WCC in the days before and after the PIEs, we execute the normalization and equalization like before.

Figure 7 shows the relative number of WCC in the days preceding and succeeding PIEs. We discover that there is an upward trend in the number of WCC. That means the followers are divided into more fragmented communities, though the followers become more similar over time. It's not hard to explain. The followers of a user increase over time since new followers arrive faster than old followers leave in ordinary times. While these new followers have parallel interests, there is no enough time for them to know each other and follow each other (they are likely to follow each other after a period of time but not immediately). Hence, there would be more weakly connected components of Twitter ego networks.

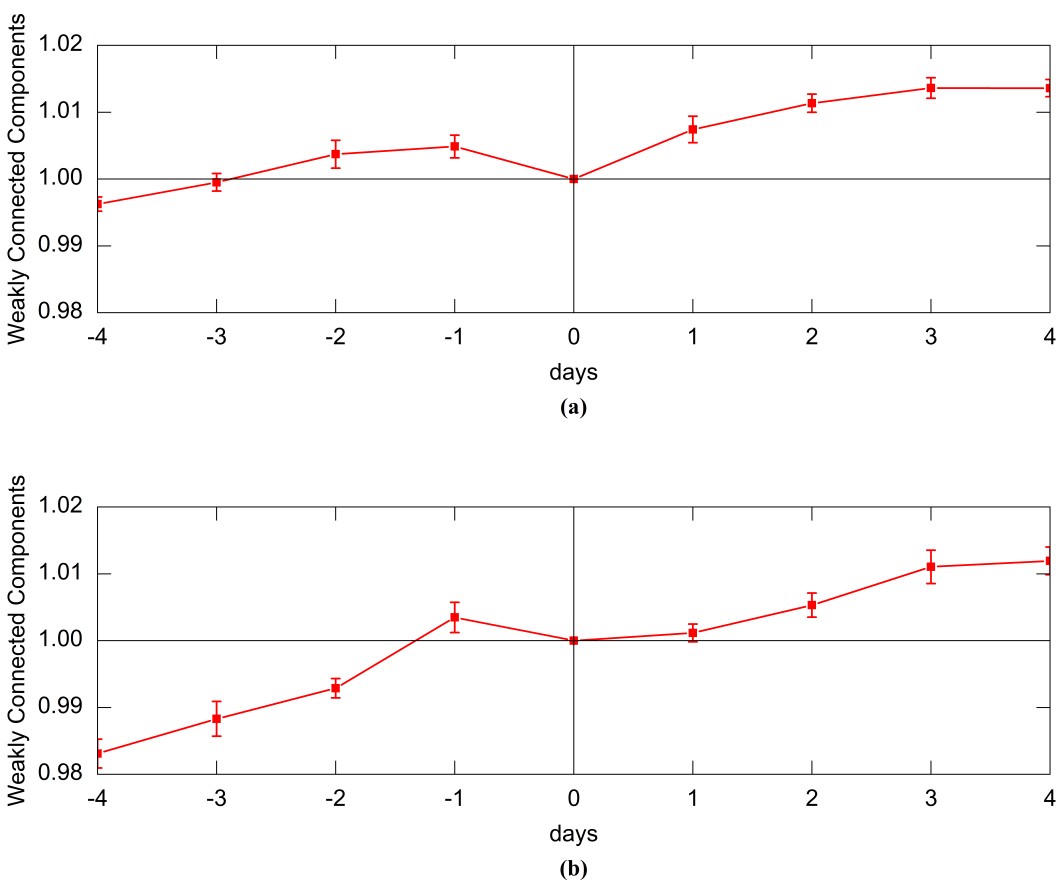

**Figure 7.** (**a**) The number of normalized Weakly Connected Components before and after a follow-burst-PIE; (**b**) The number of normalized Weakly Connected Components before and after an unfollow-burst-PIE.

This upward trend of the number of WCC is interrupted by PIEs. Whether a follow-burst-PIE or an unfollow-burst-PIE decreases the number of WCC. A PIE prompts users to post tweets to discuss the event, and this speeds up the process of users knowing each other. While a follow-burst-PIE brings numerous new members to the ego network, the newcomers may be familiar with old ones in a short time and follow each other. The number of WCC, therefore, will not increase after a follow burst-PIE. On the other hand, an unfollow-burst-PIE expels those users who are not close to the others out of

the ego network. The ego network then becomes more tightly connected and the number of WCC gets reduced.

*5.4. Followers Following Each Other*

Lastly, we focus on the edge density of the ego networks. For a given user, the metric represents what fraction of all possible following relationships between his followers actually exist. Edge density measures the degree that a user's followers tend to follow each other. A lower value of the number of WCC indicates that information can spread to a broader range, while a higher value of edge density means a faster propagation speed in a local scope. Similarly, we measure the edge density of ego networks in the days before and after the PIEs. Normalized treatment and averaging treatment are given then.

Since users' ego networks always keep growing in the number of nodes, we guess that the edge density of ego networks declines over time. Figure 8 confirms our thought. We notice that there is a steady decrease in edge density before either type of PIEs. For the days after a burst, however, something interesting happens. For the unfollow-burst-PIEs, the density increases, while for the follow- burst-PIEs, the density still decreases but more slowly. We explain the two observations as follows. A follow-burst-PIE will bring plentiful new followers to the ego. While the newcomers and old followers will follow each other within a short time, as we mentioned above, the newly-established follow connections won't be too many. Thus the edge density will still decrease but slower. On the other hand, an unfollow-burst-PIE won't bring too many new followers but expel users who are loosely connected with others. Therefore, the edge density rises after an unfollow-burst-PIE. In general, both follow-burst-PIEs and unfollow-burst-PIEs inhibit the downtrend of the edge density.

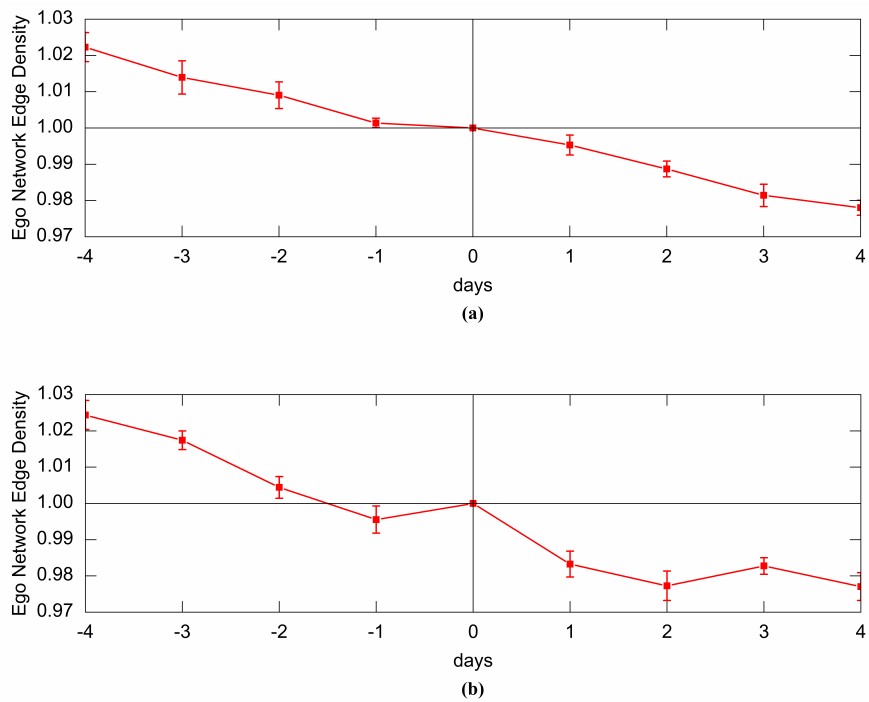

**Figure 8.** (**a**) Normalized Edge Density before and after a follow-burst-PIE; (**b**) Normalized Edge Density before and after an unfollow-burst-PIE.

## 6. Conclusions

In this paper, we propose an event detection method to detect the occurrences of significant events for specific individuals. This method only based on the variation of the user's followers is quite simple but effective. We divide the events into categories according to the positive or negative effect on the particular user. Further, we observe the evolution of individuals' Twitter followership networks

and see if different types of events have different influences on the network dynamics. On some of the features, events accelerate the original evolutionary trend. In other features, events suppress the original trend.

Understanding the evolution trend of Twitter followership networks and its reactions to events are helpful to investigate further on Twitter followership networks. We can control the occurrence of events to make target networks achieve a desired state. This can be applied in various fields such as public opinion monitoring, disaster warning, crisis management, and intelligent decision making. Besides, our work in this paper is limited to the analysis of Twitter followership networks. Further works can be done for the research of network dynamics on more kinds of Twitter networks such as retweet networks and mention networks.

**Author Contributions:** Conceptualization, T.T. and G.H.; methodology, T.T.; software, T.T.; validation, T.T.; formal analysis, T.T.; investigation, T.T.; resources, T.T.; data curation, T.T.; writing—original draft preparation, T.T.; writing—review and editing, T.T.; visualization, T.T.; supervision, G.H.; project administration, G.H.; funding acquisition, T.T. All authors have read and agreed to the published version of the manuscript.

**Funding:** This research was funded by the National Natural Science Foundation of China grant number 0561701074.

**Conflicts of Interest:** The authors declare no conflict of interest.

## Abbreviations

The following abbreviations are used in this manuscript:

| | |
|---|---|
| API | Application Program Interface |
| PIE | Personal Important Event |
| SOLR | Searching on Lecene with Replication |
| CRF | Conditional Random Field |
| VDEH | Variable Dimensional Extendible Hash |
| STSS | Space-Time Scan Statistics |
| LECM | Latent Event and Category Model |
| REST | Representational State Transfer |
| TF-IDF | Term Frequency-inverse Document Frequency |
| WCC | Weakly Connected Components |

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
