# Peer review of "Detecting and Tracking Significant Events for Individuals on Twitter by Monitoring the Evolution of Twitter Followership Networks"

_information, doi:10.3390/info11090450_

Round 1

Reviewer 1 Report

The paper is devoted to the interesting and actual topic of identification of significant events by individuals' behavior in Twitter. The authors suggest to make monitoring of the evolution of a Twitter followership networks. By detecting the significant (in some sense) changes in the network we can say that important event happened. The idea is interesting and seems to be useful not only for politics, but also for any social sciences protecting any system against wrong information spreading and other problems.

I find the paper well written but I feel some lack of mathematical and technical features in model description. My remarks and concerns are mostly about the presentation of the main results:

  1. Is the formula (1) original? If no, the reference is needed. I also think this formula is required a spectial explaination because it is significunt for analysis results.
  2. How to choose the threshold for f(ti)? What is an advise for choosing an appropriate threshold?
  3. The detected PIE represented in Table 1 with the dates can be correlated with the dates in days from Figure 1 and 2. I recommend to add the graphs where PIE1,...PIE12 are marked in the time line on the initial figures.
  4. I recommend to add the formular for computing follower tweet similarity (see line 192).
  5. The similar to the previous remark: to give a formula of computing cosine similarity.
  6. Line 231: twice "the" word.
  7. In Conclusion, it is better to describe the contribution of the authors. Is the method they suggest new in comparison with existing methods in the literature?

I recommend a minor revision of the paper according to the remarks given above.

Reviewer 2 Report

line 21 the word contents should be singular as 'content' and for subsequent references to the 'content' of the posts

line 36 'the concern people have about real...'

line38 'or which events he or she is concerned about'

line 39 'we refer to these events as P...'

For the burst detection did you not consider an established method such as https://www.cs.cornell.edu/home/kleinber/bhs.pdf Kleinberg, J. (2003). Bursty and hierarchical structure in streams. Data Mining and Knowledge Discovery, 7(4), 373-397. I do not see where that is discussed in section 4.2

In 5.1, is the similarity using the hashtags as well?

5.2 How would the pair similarity calculation scale? There are many pairs to calculate and also if there is a stream requiring the update to be made.

For 5.2 would the normalization not reduce the detection capability for the burst? Is it enough to determine the burst from a change in ratios which can be sensitive to low changing periods?

How does the graph in 5.4 affect the results from 5.2? It is unclear how these different metrics fit together.

What is actually novel? It appears as if there are standard measures applied to a subset of the data and some basic plots are produced? Is there any novel inference or modeling approach?

It would be advisable to also look at another dataset and provide a more extensive exploration to be more convincing.

Round 2

Reviewer 2 Report

My issues raised have been addressed